# *Salmonella* as a Promising Curative Tool against Cancer

**DOI:** 10.3390/pharmaceutics14102100

**Published:** 2022-10-01

**Authors:** Ram Prasad Aganja, Chandran Sivasankar, Amal Senevirathne, John Hwa Lee

**Affiliations:** 1Department of Veterinary Public Health, College of Veterinary Medicine, Jeonbuk National University, Iksan 54596, Korea; 2Department of Preventive Medicine, College of Veterinary Medicine, Chungnam National University, Daejeon 34134, Korea

**Keywords:** *Salmonella*, *Salmonella*-mediated cancer therapy, combination therapy, anti-tumor mechanism

## Abstract

Bacteria-mediated cancer therapy has become a topic of interest under the broad umbrella of oncotherapy. Among many bacterial species, *Salmonella* remains at the forefront due to its ability to localize and proliferate inside tumor microenvironments and often suppress tumor growth. *Salmonella* Typhimurium is one of the most promising mediators, with engineering plasticity and cancer specificity. It can be used to deliver toxins that induce cell death in cancer cells specifically, and also as a cancer-specific instrument for immunotherapy by delivering tumor antigens and exposing the tumor environment to the host immune system. *Salmonella* can be used to deliver prodrug converting enzymes unambiguously against cancer. Though positive responses in *Salmonella**-*mediated cancer treatments are still at a preliminary level, they have paved the way for developing combinatorial therapy with conventional chemotherapy, radiotherapy, and surgery, and can be used synergistically to combat multi-drug resistant and higher-stage cancers. With this background, *Salmonella**-*mediated cancer therapy was approved for clinical trials by U.S. Food and Drug Administration, but the results were not satisfactory and more pre-clinical investigation is needed. This review summarizes the recent advancements in *Salmonella**-*mediated oncotherapy in the fight against cancer. The present article emphasizes the demand for *Salmonella* mutants with high stringency toward cancer and with amenable elements of safety by virulence deletions.

## 1. Introduction

Cancer is a leading cause of death worldwide and a burgeoning health burden with a limited number of successful therapeutics. On average, 10 million people worldwide lose their lives annually due to various cancers [1]. Every cancer requires an accurate diagnosis and prompt treatment at the earliest possibility. Even though most conventional treatment strategies such as surgery, chemotherapy, and radiotherapy remain major life savers, they have serious limitations that can damage healthy tissues [2,3]. Surgical removal of cancers can be effective in certain types and developmental stages of cancers; however, cancer relapse and the possibility of further spread due to metastasis are some of the inherent weaknesses of this method [4]. On the contrary, radiotherapy and chemotherapy provide varying degrees of success and inflict unprecedented failures in cancer treatment, especially distant tumor recurrences and undesirable effects [5,6]. Hence, to fill the gap, novel treatment concepts and strategies are essential as an ideal treatment for cancers. Cancer tumors consist of hypoxic core regions and necrotic centers, which make most cancer treatments incompetent due to lack of oxygen and abnormal vasculature. Studies have demonstrated that such regions are the key features of tumors that lead to treatment failure [7,8,9]. In addition, due to the abnormal vascular architecture, it is a huge challenge to deliver therapeutic agents to the tumor region. Hence, it is evident that a single treatment strategy may not be effective against cancer malignancies, but holistic approaches might bring suboptimal outcomes.

In recent decades, bacteria-mediated cancer treatments (BMCT) have garnered attention as an alternative strategy to treat cancer tumors due to the intrinsic challenges of conventional cancer treatment strategies. Advancements in genetic engineering and recombinant DNA technology had paved the path for developing numerous bacterial strains as model systems to be used in cancer immunotherapy. William Coley’s controversial study, more than a century ago, revealed that some bacterial species may hold the key to creating targeted treatments for cancers that are challenging to cure. According to Coley, the complex cocktail he produced had the potential to shrink cancer tumors; however, the lack of progressive techniques and poor understanding of the mode of action made it difficult to reproduce consistent results. Revitalizing these early findings, scientists around the world have attempted to use novel bacterial species, such as *Bifidobacterium*, *Clostridium*, *Salmonella*, *Streptococcus*, and *Listeria monocytogenes* for tumor regression and have brought deep insight into their mode of action.

Accumulating evidence suggests that the cytotoxic effect of several bacteria can be hijacked and used against cancer cells. Some bacteria are naturally capable of homing to tumors, while other invasive species have been exploited to deliver heterologous genes intracellularly. The high rate of replication and invasive characteristics of some species, including *Salmonella* and *Clostridium*, are better vector candidates to express the target gene in tumor cells. Bacteria that have higher penetration and dispersion throughout tumors hold a greater ability to regress tumors [10]. Tumors have hypoxic regions where facultative anaerobes can survive and function to produce anti-tumor effects. Efficient delivery of the therapeutic gene to the target tissue or cell is the most significant hurdle for successful gene therapy. Thus, DNA is normally combined with a gene delivery vehicle in order to protect and mediate the effective tissue or cell entry of the payload gene of interest (GOI). The account for the payload GOI is handled by the bacterial invasive character that stands as a model in animal and human infection.

Among known bacterial species suitable for BMCT, *Salmonella enterica* serovar Typhimurium (ST) is one of the most versatile, as it can grow under both aerobic and anaerobic culture conditions. Therefore, it has no problem spreading systemically in animals under highly aerobic conditions, and eventually localizes in hypoxic tumor regions, which are their preferred sites of colonization. *Salmonella* has exhibited an immense capability to colonize hypoxic, necrotic, and metastatic tumors; thus, it can complement conventional treatment strategies [11]. The preferential accumulation ratio of ST in tumor regions is between 10^3^ to 10^4^ times more than in normal body tissues, resolving a huge challenge in the target specificity of cancer treatment [12]. Experimental data have confirmed that this localization occurs in a few days in most cases [13]. Thus, *Salmonella* can be used as a carrier to deliver therapeutics directly into the tumor regions, thereby protecting them from degradation and potential damage from the host immune system. Moreover, *Salmonella* can be easily tailored in a myriad of ways such as bacterial ghost systems, protein secretion systems, target-oriented and lysis systems, quorum sensing systems, etc. Thus, it has versatility for cancer treatment. Additionally, the simplicity of manufacturing it, its cost-effectiveness, and its rapid mass production place it as a new option for cancer therapy. In the present article, we elaborate on the mechanisms, strategies, and potential engineering methods of *Salmonella* as a key microbial agent suitable for the next generation of cancer therapy.

## 2. Bacterial Application for Cancer Therapy

Numerous studies have demonstrated the anti-tumor effects of several bacteria, either by directly killing or modulating immune components of the tumor microenvironment. The natural cytotoxic features of bacteria can result in substantial tumor regression. Therefore, many researchers have exploited non-pathogenic obligate anaerobes and facultative anaerobes which selectively infiltrate and replicate within solid tumors when administered systemically. The ability of bacteria to regress tumors came into the limelight in the early 1800s. BMCT started gaining momentum when Coley’s toxin, a mixture of killed *Streptococcus pyogenes* and *Serratia marcescens* developed by Dr. William B. Coley, achieved clinical responses for many malignant tumors [14]. Additionally, the anti-tumor characteristics of various bacteria have been documented, such as *Salmonella* [15], *Escherichia coli*, *Vibrio cholerae* and *Listeria monocytogenes* [16], *Clostridium welchii* [17], *Clostridium tetani* [18], *Bifidobacterium infantis* [19], *Streptococcus pyogenes* [20], and *Proteus mirabilis* [21]. *Salmonella* holds natural cytotoxicity that regresses tumors when injected in its native form [15]. Similarly, *Clostridium*, an obligate anaerobe, can regress tumors in mice [17,22,23].

The bacterial implication in cancer can function as a two-edged sword, since the association of certain pathogenic species has been linked to colon cancer. *E. coli* possesses a genomic island polyketide synthetase codes for the synthesis of colibactin that has been implicated in colorectal cancer [24]. In another scenario, *Clostridium* sps., especially *Clostridium perfringens* and *Clostridium septicum*, has been associated with colorectal cancer [25,26]. *Salmonella typhi/p*aratyphi produces a potent carcinogen N-nitroso compound and has been documented for hepatobiliary carcinoma [27].

## 3. Attenuated Bacteria for Cancer Therapy

The unique tumor-homing characteristics of bacteria have been exploited for anti-tumor vaccine therapies, and such bacteria can be genetically attenuated to carry and deliver heterogeneous antigens which elicit host immunity. Generally, attenuation by modification on lipopolysaccharides, flagella, or other structural proteins is widely practiced. Moreover, with advancements in molecular biology, genetic modification for the delivery of different agents, such as tumor-associated antigens, immunostimulatory molecules, anti-tumor drugs, and nucleotides (DNA or RNA), are extensively accepted approaches [28]. A bacterial platform targeting cancer therapy is the choice of many researchers, as it possesses many benefits. Genetic manipulation, precise tuning, and limitless functional combinations of desired characteristics using bacteria have brought cancer treatment to a new horizon. Recently, several studies have focused on cancer treatment using the direct delivery of heterogeneous antigens or genes encoding anti-tumor molecules. Bacteria, including *Salmonella* species, have been significantly attenuated in their virulence in order to rule out a major threat of systemic infection or collateral damage [13,29].

Bacteria can specifically target tumors and actively penetrate tissue, which can induce cytotoxicity and destroy malignant cells [30]. Obligate anaerobes and facultative anaerobes can have an intrinsic tumor-targeting ability, because they can survive intra-tumor hypoxia. The hypoxic area in the tumor facilitates anaerobes and facultative anaerobes, as well as additional nutrients, including purines, and the immunosuppressive environment inhibits the clearance of *Salmonella*. Transforming growth factor-beta within a tumor inhibits the activation of neutrophils that prevent bacterial clearance from solid tumors [31]. Attenuated *Salmonella* strains which have been developed as gene delivery vectors carrying herpes simplex virus thymidine kinase (HSV TK) possess anti-tumor activity in mice and are capable of both selective amplification within tumors and expression of effector genes encoding therapeutic proteins [15].

Several strategies have been exploited for BMCT research, such as bacterial tumor-targeting properties, intra-tumoral penetration, bacterial cytotoxicity, expression of anti-cancer agents, host gene-triggering strategies, and immunotherapy. Although efforts are ongoing, the success of BMCT in animal models could not be replicated in humans using ST VNP20009. The attenuated strain with chromosomal deletion of the *purI* and *msbB* genes was able to target the tumor and inhibit tumor growth in mice [32], but failed to replicate anti-tumor effects in humans [33].

## 4. Why Is *Salmonella* the Best Option?

*Salmonella*-mediated cancer therapy (SMCT) is gaining popularity over other therapies because of several advantages. For instance, it has characteristic features that are peculiar to the species, such as self-targeting tumor localization and proliferation, and intrinsic anti-tumor nature. Compared with other bacterial species, it has several advantages, such as the relative ease of attenuation and gene manipulation, and it can grow in a hypoxic environment. It has a wide range of hosts, including humans and farm animals. In addition, it can be administered orally and stimulate local and systemic immune responses, highlighting its use as a model vector for cancer vaccine therapy.

### 4.1. High Tumor Colonization

The inherent facultative anaerobic and intracellular pathogenic characteristics of *Salmonella* facilitate strong preferential colonization in tumor tissue (Figure 1). Clairmont et al. have documented 1000-fold accumulation of the ST VNP20009 strain in tumors compared to the liver [32]. These attenuated strains were cleared rapidly from systemic circulation, liver, and spleen, whereas proliferation in tumor tissue continued for a longer time [32]. Thus, lower toxicity is observed in the host. Such selective tumor colonization and proliferation are owing to the hypoxic and vascularized environment of the tumor [34]. The mobility of *Salmonella* toward tumors and their accumulation is influenced by the tumor microenvironment, host reticuloendothelial system, and bacterial metabolism [35].

### 4.2. Non-Specific Tumor Target

*Salmonella* has been reported as having invasiveness toward a broad spectrum of murine tumor models. Several documents support its effectiveness in the treatment of melanoma, colon cancer, lung cancer, prostate cancer, cervical cancer, and metastatic T-cell lymphoma. The quiescent cells in the tumor cylindroid model secrete bacterial chemoattractant, and the presence of necrotic and quiescent cells attracts and enables ST to replicate in the central regions of the quiescent tumor cell mass [36]. Other mechanisms revealed that *Salmonella* uses aspartate receptors to initiate chemotaxis toward tumor cylindroids in vitro. It uses the serine receptor to initiate penetration and the ribose/galactose receptor to enhance affinity toward necrotic tissue [37].

### 4.3. Inherent Anti-Tumor Character

*Salmonella* is believed to have an intrinsic oncolytic activity that is mediated through the induction of tumor cell apoptosis. Bacterial invasion releases toxins, deprives the tumor microenvironment of nutrients, and promotes apoptosis [30]. Alternatively, the anti-angiogenic ability of *Salmonella* could delay tumor progression by inhibiting tumor angiogenesis [38,39]. Findings on the ability of *Salmonella* to downregulate the expression of hypoxia-inducible factor-1 alpha (HIF-1α) and vascular endothelial growth factor (VEGF) through the AKT/mTOR pathway support the anti-angiogenic ability of *Salmonella* [38]. Furthermore, bacteria can activate innate immune cells by cytokine stimulation and recruit and activate other immune cells at the tumor site, promoting anti-tumor immunity [40].

### 4.4. Engineering Plasticity

*Salmonella* is an excellent system for acquiring attenuation by targeting different approaches. It has flexibility in gene modification that helps to improve the safety of bacterial therapy through the deletion of major virulence factors or the generation of auxotrophic mutants that are incapable of replicating efficiently in an environment deficient in specific nutrients [41]. This increased specificity allows it to function as a “bacterial robot” to deliver anti-tumor therapeutic agents to the tumor site. It can deliver therapeutic payloads in the form of DNA, RNA, or protein. For this, the *Salmonella* delivery system needs to be transformed with extrinsic gene expression plasmids. The engineering plasticity of *Salmonella* enhances its efficacy in cancer therapy through improving safety, specificity, and delivery of anti-tumor therapeutic agents. It has been exploited for the delivery of cytotoxic agents such as cytolysin A, PE38, and diphtheria toxin, apoptosis-inducing proteins such as Fas ligand and TNF-related, apoptosis-inducing ligand (TRAIL), as well as apoptin and immunomodulatory cytokines, including IL-2, IL-12, IL-18, and IFN-γ [34,42,43,44,45,46]. Engineering plasticity enables the application of *Salmonella* in the expression of anti-cancer agents, “prodrugs,” or tumor-specific antigens at tumor sites. At present, the expression of oncogene-silencing RNA and a strategy of incorporating tumor-killing nanoparticles are being developed [30,47].

Recently, protein drug delivery to tumors was demonstrated by Raman et al. The group used intracellular delivery of *Salmonella* to express constitutive two-chain active caspase-3, an engineered form of caspase-3 that causes apoptotic cell death. They developed the strain by engineering genetic circuits to enable autonomous deposition of protein payloads directly into cancer cells that decreased tumor growth and reduced breast metastases [48]. Transforming growth factor alpha (TGFα) is a natural ligand for epidermal growth factor receptor (EGFR), a receptor highly expressed in tumor cells [49]. Researchers employed the ΔppGpp *Salmonella* mutant to deliver a recombinant drug TGFα-PE38, an immunotoxin comprising a modified *Pseudomonas* exotoxin A (PE38) conjugated with TGFα, and investigated the process through which it undermined the tumor by inducing the expression of pro-inflammatory cytokines from macrophages and neutrophils, such as IL-1β and TNFα [46,50]. The recombinant protein produced by bacteria effectively regressed the solid tumor growth and induced tumor cell apoptosis [46].

## 5. *Salmonella enterica* Serovar Typhimurium

ST, a gram-negative pathogenic bacteria [51], is one of the most extensively studied and promising bacterial mediators of cancer immunotherapy. ST is the choice of many researchers because of its ability to grow in both aerobic and anaerobic environments, resulting in its colonization in both non-hypoxic and hypoxic tumors [52]. It has high tumor specificity and deep tissue penetration potential. It possesses chemoreceptors, such as the aspartate receptor, that initiate chemotaxis toward viable tumor tissue, serine receptors to induce tissue penetration, and a ribose receptor to regulate migration toward necrotic tissue [37]. In addition, it has been engineered and designed in many studies that have explored cancer-targeting therapeutic agents (Table 1). ST has been engineered as a delivery vector for anti-tumor effects by expressing the pro-apoptotic Fas ligand in breast cancer and colon carcinomas in mice models [53]. TRAIL, a natural inducer of apoptosis and tumor cell death, was used in ST-based cancer therapy under the control of a prokaryotic radiation-inducible promoter, recA. This study has shown the inhibition of mammary tumor growth and substantially increased rates of survival [54]. 

Since the recognition that bacteria could be used for cancer therapy in the 19th century, *Salmonella* has been widely studied. It has been the best option for researchers as it covers many features, including high tumor-targeting capacity, tumor specificity, deep tissue penetration, and engineering plasticity. ST with auxotrophic mutations has demonstrated prominent anti-cancer potential [13,15,67,68]. Attenuated *Salmonella* with modification on lipopolysaccharide has been reported to have an anti-tumor effect [69]. ST with a modified lipid A (ST VNP20009) has achieved anti-tumor responses in dogs [70]. Lipid A-modified ST replicated more than 1000 times in tumors compared to normal tissue [68]. Auxotrophs for aromatic compounds, such as tryptophan [13] and purine, revealed *Salmonella* to be a good candidate for cancer therapeutics.

## 6. ST VNP20009 Strain

Several ST mutants have been evaluated for cancer therapy. The strains that are studied most frequently are ST VNP20009, A1-R, and other mutants. Among them, the ST VNP20009 strain was generated to obtain stable attenuated virulence by deletion of the *purl* gene for purine auxotrophy, and endotoxicity was reduced by the deletion of the *msbB* gene with a lipid-A modification [32]. ST VNP20009 renders selective accumulation at a ratio of >1000:1 over normal organs and specificity toward the tumor tissue, and is highly attenuated with a virulence reduction of about 10,000-fold [12]. This strain has been used in many studies and is the only strain to be evaluated in phase I clinical trials for metastatic melanoma or renal cell carcinoma in humans [33,71]. Mice were implanted with murine melanoma, and tumor growth was inhibited in the human tumor xenografts by 57–95% with a single dose of ST VNP20009 [72]. Aside from solid tumors, ST VNP20009 can induce apoptosis in multiple types of leukemia cells and prolong the survival of the MLL-AF9-induced acute myeloid leukemia-carrying mice [73]. Zheng et al. have demonstrated transgenic expressions of proteins by ST VNP20009 [74], and it has been used for the delivery of cycle-inhibiting factor genes for the treatment of colon cancer [75]. The lack of tumor regression in clinical trials of ST VNP20009 [33] compelled further modification, and the *phoP/phoQ* system was deleted in order to enhance tumor-targeting ability [76]. Complete genome sequencing of ST VNP2009 revealed nonsynonymous single nucleotide polymorphisms and *purM* deletion, demanding further study for effective human application [77].

A new method for tumor localization and improved safety has been demonstrated by coating VNP20009 with tumor cell-derived nanoshells [78]. The carcinoma cell-mimetic bacteria (CCMB) were released from carcinoma cells using UV irradiation, with an extra membrane from apoptotic bodies of invaded tumor cells as coating shells. The presence of a tumor cell-derived membrane in CCMB elicits low inflammation and enhances homologous-targeting tumor localization. This strategy exhibited tumor regression and metastasis inhibition and stands as a promising biotherapy for tumor treatment [78].

## 7. ST A1-R Strain

ST A1-R strain is a tumor-seeking leucine-arginine auxotrophic mutant developed by nitrosoguanidine mutagenesis that grows in viable as well as necrotic regions of tumors, but is restricted to normal tissue. It has been characterized to treat various tumor models in mice such as prostate, breast, pancreatic, and ovarian cancer, as well as sarcoma and glioma [79]. Mice implanted with metastatic PC-3 human prostate tumors were cured by nearly 70% when injected weekly with ST A1-R [80]. Likewise, mouse models with disseminated and metastatic human ovarian cancer cell line SKOV3-FGP demonstrated remarkable tumor size reduction and survival compared to untreated mice [81]. Administration of ST A1-R to nude mice with primary osteosarcoma and lung metastasis was highly effective, especially against metastasis [79]. Bone metastasis is a lethal and morbid late stage of breast cancer. ST A1-R treatment reduced bone growth in highly metastatic human breast cancer in nude mice, and also helped to prevent and inhibit breast cancer bone metastasis [82]. This strain also has the capability of decoying quiescent cancer cells to the S/G2/M phase and sensitizing them to cytotoxic chemotherapy [83].

## 8. Other Auxotrophic Mutant Strains

Auxotrophic mutant strains are an attractive alternative modulation of *Salmonella* for killing tumor cells. We engineered ST auxotrophic for tryptophan as a candidate for cancer therapy. This engineered mutant has improved the ability to target and colonize the tumors, reduced the fitness in healthy tissues, and inhibited primary tumor growth and lung metastases [13]. Liang et al. modified the auxotrophic *Salmonella* vector harboring ∆*aroA* and ∆*purM* mutations to deliver antitumor molecules, including the angiogenesis inhibitor endostatin and apoptosis inducer TRAIL, then evaluated anti-tumor efficacy. This strain significantly suppressed tumor growth and prolonged the survival of colon carcinoma and melanoma-bearing mice [84]. Attenuated ST *aroA* strains that secrete prostate-specific antigens in combination with cholera toxin subunit B induced cytotoxic CD8+ T cells, which efficiently prevented tumor growth in mice [85]. Another application of *aroA* deleted *ST* (*Δdam*, *ΔaroA*) demonstrated the suppression of tumor angiogenesis, tumor growth, and metastasis while delivering a murine MHC class I antigen epitopes of Legumain, a protein highly expressed in tumor-associated macrophage (TAM) [86].

## 9. Anti-Tumor Mechanism of *Salmonella*

The anti-cancer activities of *Salmonella* have been well documented, but the details of killing tumor cells are still under debate, and accumulated evidence supports the different mechanisms proposed. Various intrinsic features of *Salmonella* and the tumor microenvironment play a crucial role during the proliferation, progression, and metastasis of tumor cells (Figure 2).

### 9.1. Hypoxic Environment and Tumor Vasculature

Chronic and acute hypoxia occurs in the tumor due to unevenly distributed and chaotic tumor vasculature, which is associated with decreased oxygen supply and disturbed cell proliferation. A hypoxic environment favors facultative anaerobic bacteria such as *Salmonella*. Tumor angiogenesis can cause an abnormal vasculature that is more vascularized than the corresponding normal tissue [102]. Such a high and unruly vascularizing tumor microenvironment enhances the chance of *Salmonella* invasion. Higher localization of such facultative anaerobic bacteria starts killing tumor cells for their nutrients and survival, which is further supported by chaotic vasculature. A positive correlation between higher vascularity and destruction of tumor blood vessels by the tumor-targeting double-auxotrophic mutant ST A1-R has been demonstrated [39].

### 9.2. Abundant Nutrients and Competitive Nature of Salmonella

The tumor microenvironment hosts diverse micronutrients. A study in murine pancreatic and lung adenocarcinoma models revealed that nutrients available to tumors differ from those present in circulation that influence cancer cell metabolism [87]. Metabolic adaptation in tumors accumulates metabolites that serve as a substrate for the generation of energy and biomass, as well as altering normal gene expression [103]. This creates an immunosuppressive microenvironment inside the tumor that could ease the survival and growth of attenuated auxotrophic bacteria by supplying nutrients and providing protection from the immuno-surveillance of the host [104]. By using this knowledge, auxotrophic *Salmonella* for leucine, arginine, and tryptophan has been exploited for the treatment of prostate cancer and breast cancer in mice models [13,105].

### 9.3. Tumor Penetration

The potential for deep tumor penetration has made *Salmonella* the best candidate for BMCT. Conventional chemotherapeutic drug distribution relies on passive transport, limiting uniform delivery in the inner tumor region. Live *Salmonella* gathers energy from surrounding abundant nutrients and invades deep tumor tissue. After systemic administration of ST, it is dispersed in all the regions of the solid tumor from the edge to the core center, and apoptosis is induced [99]. Such a spatial distribution of bacteria within tumor tissue is influenced by bacterial motility; bacteria with higher motility penetrate deeper [106]. Other groups believe that ST migration in the tumor is a passive process that is independent of motility and chemotaxis, however, they have used different strains and time points post-infection to examine the tumor-colonization events [35]. Other factors including host immune response can determine tumor penetration. Neutrophils prevent the spread of bacteria from the necrotic region into tumor tissue, and their depletion increases intra-tumor bacterial colonization [107].

### 9.4. Apoptosis and Autophagy-Inducing Intrinsic Anti-Tumor Action

Numerous investigations have shown that *Salmonella* has direct killing effects on cancer cells. Accumulation of *Salmonella* in tumors induces apoptosis [99]. High-resolution multiphoton tomographic images have revealed that genetically engineered ST A1-R infected cancer cells expanded, burst, and finally lost viability [101]. *Salmonella* has an intrinsic ability to kill cancer cells by inducing both cellular apoptosis and autophagy [100,108]. Although the exact mechanism underlying the induction of apoptosis by *Salmonella* is less defined, competition for nutrients with cancer cells and the release of bacterial toxins may induce apoptosis. It could also induce autophagy, a scavenger process that is present at a lower level in tumor cells than in their normal counterparts. *Salmonella* induces the autophagic signaling pathway in a dose- and time-dependent manner via downregulation of the protein kinase B (AKT)/mammalian target of the rapamycin (mTOR) pathway [109]. The AKT/mTOR pathway plays a significant role in cellular physiology and homeostasis. Downregulation of this pathway reduces the expression of matrix metalloproteinase 9 (MMP-9), an oncoprotein involved in metastasis [110].

### 9.5. Inhibition of Angiogenesis

Angiogenesis plays a vital role in the development and progression of tumors. HIF-1α and VEGF play a significant role in tumor angiogenesis. *Salmonella* invasion in a tumor can downregulate the expression of HIF-1α and VEGF, and inhibits tumor angiogenesis via the AKT/mTOR pathway [38]. Another mechanism for the suppression of angiogenesis involves a recently identified tumor angiogenesis inhibitor protein, connexin 43 (Cx43), which functions by downregulating VEGF via HIF-1α [111]. Furthermore, higher tumor vasculature has been correlated with the destruction of tumor blood vessels with the tumor-targeting double-auxotrophic mutant ST A1-R [39,112].

### 9.6. Immunomodulation in Tumor Tissue

*Salmonella* can manipulate immune components of the tumor niche in favor of tumor inhibition by shifting the tumor microenvironment from immunosuppressive to immunogenic (Figure 1). This process involves alterations to both cellular and soluble components of the immune system which, in turn, affects the phenotypic and functional properties of immune cells. *Salmonella* infection has been reported to increase the infiltration of macrophages [88], natural killer (NK) cells, CD4+ helper T cells, CD8+ cytotoxic T cells [89], and B cells [90]. Tumor cells release chemo-attractants such as colony-stimulating factor 1 (CSF-1) and the chemokine C-C motif chemokine ligand 2 (CCL2) to recruit monocytes that differentiate into M2 macrophages [113]. M2 macrophage polarization supports the growth and malignancy of tumors by suppressing the anti-tumor immune responses of the host by secreting immunosuppressive molecules, such as arginase 1 (Arg1) [96] and the cytokine IL-10 [114]. Upon *Salmonella* invasion, TAMs increase the expression of M1 macrophage activation markers, such as the IFN-γ-dependent Sca-1 and MHC class II proteins [88]. This is a paradigm for shifting M2 to M1. These M1 macrophages orchestrate protective anti-tumor immune responses through the expression of nitric oxide synthase (NOS2) and TNF-α [98].

Regulatory T (Treg) cells mitigate anti-tumor immunity by inhibiting tumor antigen-specific cytotoxic T-lymphocytes (CTL) [115]. Intra-tumoral injection of attenuated *Salmonella* has immunotherapeutic potential, as it reduces Treg cells in a colon cancer model [93]. *Salmonella* treatment downregulates the cell surface molecule, CD44 present in both Treg and cancer cells [94]. Cx43 is a ubiquitous protein for gap junction formation but is normally lost during melanoma progression. It has been reported to suppress the growth of melanoma and is involved in peptide transfer to activate T cells and induce anti-tumor immunity [116]. *Salmonella* infection can induce the upregulation of Cx43 and facilitate gap junction formation. Dendritic cells can collect peptide antigens via gap junctions and present antigenic peptides from the tumor cells. Antigenic presentation activates cytotoxic T cells against the tumor antigen, which can ultimately control the growth of distant uninfected tumors [92].

### 9.7. Orchestration of TAM Function and Polarization

TAMs are a major component of the leukocytic infiltrate of tumors, and have served as a paradigm for cancer-related inflammation, promotion tumor growth, invasion, metastasis, and drug resistance [117]. TAM has an M2-like phenotype, although there is a state of constant transition between the two forms (M1 and M2) [118]. M1 macrophages are well-known for anti-tumor functions, whereas the M2 macrophages promote the occurrence and metastasis of tumor cells, tumor angiogenesis, and tumor progression [119]. TAM functions as a double-edged sword, with the ability to express pro- and anti-tumor activity, and has excellent reprogramming potential toward immunological stimuli such as IFN-γ or IFN-α. Upon bacterial invasion, TAM is associated with an innate response compelled to produce inflammatory cytokines. This stimulation in TAM re-programs immunosuppressive M2 macrophages into immunostimulatory M1 macrophages [120,121]. *Salmonella* invasion increases high mobility group box 1 (HMGB1) secretion in tumors to coax the polarization of macrophages in favor of an M1-like phenotype, which is reflected by increased inducible nitric oxide synthase (iNOS) and interleukin 1 Beta (IL-1β) production [122]. In a separate investigation, the M2 macrophage phenotype was reversed into the M1 macrophage in a co-culture of M2 macrophage with a genetically engineered ST YB1 strain. This suggests that ST redirects M2:M1 macrophage transition in TAM by switching the macrophage from the CD206^high^/HLA-DR^low^ phenotype to the CD206^low^/HLA-DR^high^ phenotype in order to undermine breast tumor growth [123]. Thus, generated M1 macrophages mediate cytotoxicity by releasing tumor-killing molecules ROS and NO [124], as well as by antibody-dependent, cell-mediated cytotoxicity [125]. This event orchestrates the tumor microenvironment toward protective anti-tumor immune responses.

In a different approach, suppression of TAMs can be achieved through designing cancer vaccines against proteins which is overexpress by TAMs, such as Legumain and others. Legumain encodes an asparaginyl endopeptidase that is highly upregulated in murine and human tumor tissues [86]. Administration of Legumain or its MHC class I antigen epitopes as a minigene vaccine through *Salmonella* induces specific CD8+ T-cell response and suppresses tumor angiogenesis, tumor growth, and metastasis [86,126].

### 9.8. Release of Cytotoxic Chemicals

The release of cytotoxic compounds such as perforin and granzyme is a defense mechanism during *Salmonella* treatment which helps to kill cancer cells. The expression of immunomodulatory molecules, such as cytokines and chemokines, stimulates the host immune system to clear tumors. As a consequence, *Salmonella* has been explored as a vector for the expression of cytotoxic agents such as cytolysin A, Fas ligand (FASL), TNF-α, TRAIL, IL-2, and IL-18 [53,54,66,127,128]. ST engineered to express the proapoptotic cytokine FASL inhibited the growth of primary tumors by an average of 59% for breast tumors and 82% for colon tumors [53]. Likewise, cytotoxic protein (HlyE) was expressed in the hypoxic regions of murine mammary tumors with the aid of a highly hypoxia-inducible promoter that increased tumor necrosis and reduced tumor growth [65]. ST engineered for the secretion of murine TRAIL under the control of the prokaryotic radiation inducible RecA promoter, activated apoptosis, delayed mammary tumor growth, and reduced mortality by 76% [54]. Chen et al. demonstrated similar results for suppressing melanoma by TRAIL under the control of the hypoxia-induced *nirB* promoter [63].

### 9.9. Role of the Type III Secretion System

*Salmonella* harbors a well known virulence factor, the type III secretion system (T3SS) encoded by “*Salmonella* pathogenicity island 1” (SPI1) to facilitate direct injection of effector proteins into host cells, interfere with intracellular signaling pathways, manipulate host cell cytoskeleton network, and enable colonization [129]. They are adapted to produce and store most of the effector proteins to secrete through the T3SS into the host cells [130]. This powerful system can be manipulated in order to translocate heterogeneous antigens by fusion with effector proteins. Cellular invasion promoted by SPI1 is supported by injecting bacterial effector molecules encoded by *Salmonella* pathogenicity island 2 (SPI2), which attempt to restore themselves inside the host by restructuring endosomes into *Salmonella*-containing vacuoles (SCVs) to evade host defense mechanisms and enable intracellular survival [131,132]. Thus, the invasion of the native form or bacterial delivery system is facilitated by the combined function of SPI1 and SPI2. Thus, the established *Salmonella* can induce apoptosis, and it has been supported by the apoptotic tumor cell death through the expression of *Salmonella* effectors such as a virulence factor SpvB (an ADP-ribosyl transferase enzyme) through the activation of caspases 3 and 7 [133]. *Salmonella* effector SipA has been reported to downregulate the expression of P-glycoprotein which is highly expressed in several cancers such as colon, breast, kidney, and lymphoma [134,135].

Regions of T3SS proteins such as needles, inner rod proteins, and flagellins are perceived by nucleotide-binding oligomerization domain (NOD)-like receptor (NLR) family apoptosis inhibitory protein (NAIP) to activate the NAIP-NLR family caspase-associated recruitment domain-containing protein 4 (NLRC4). This results in NLRC4 inflammasome assembly and triggers the cleavage and activation of caspase-1 and caspase-8. Activation of caspase-1 mediates pyroptosis and stimulation of caspase-8 triggers caspase-3/7 promoting apoptosis that ultimately results in inflammatory death of host cells termed as PANoptosis [136,137,138]. Internalization of effector proteins modulates the host immune response and secretes proinflammatory cytokines or rearranges the host cell cytoskeleton, facilitating bacterial invasion [91,129]. In order to strengthen the antigen delivery for cancer vaccine development, *Salmonella* T3SS is a promising strategy. It was demonstrated that the delivery of the NY-ESO-1 tumor antigen through T3SS elicited the regression of established NY-ESO-1-expressing tumors in mice [139]. T3SS from *Salmonella* has been exploited in order to translocate heterologous antigens MHC class I-peptide p60 (217–225) from *Listeria monocytogenes* into the cytosol of host cells, which induced antigen-specific CD8 T cells, for significant tumor regression [140]. SPI2 effector protein SseF was used to deliver tumor-associated antigens (TAAs) survivin into the cytosol of antigen-presenting cells for optimal immunogenicity [141]. Likewise, another SPI2 effector protein, SspH2, was used to produce a chimeric construct with the p60 protein of *Listeria monocytogenes,* which induces a p60-specific CD4 and CD8 T-cell response in vaccinated mice [142]. The assisted delivery of TAA via SPI-2-regulated T3SS produced anti-tumor activity in a mouse model.

## 10. Current Approach with Combination Therapy

*Salmonella* has been used with other therapeutic agents to enhance the efficacy of anti-cancer activities. It has been used in combination with chemotherapy, radiotherapy, immune checkpoint inhibitors, and immunomodulatory cytokines. The combined administration of *Salmonella* with chemotherapy reduces toxicity compared with individual therapy with bacteria or chemotherapeutics. For this, a murine melanoma model was treated with VNP20009 and cyclophosphamide, which induced a significant decrease in microvessel density and serum VEGF levels compared with either treatment alone. [143]. Similarly, the combination of anti-angiogenic agent HM-3 (a polypeptide inhibiting angiogenesis) and VNP20009 harboring expression plasmids for siRNA targeting *Sox2* demonstrated efficient treatment for lung cancer [144]. Another well known ST A1-R strain was implemented in combination with the chemotherapeutic drugs temozolomide, doxorubicin, and anti-angiogenic agents, which significantly suppressed the growth of tumors in patient-derived orthotopic xenograft models [59,60,145]. The co-administration of radiotherapy and BMCT produced prominent anti-tumor effects compared to either of the treatments alone. The combination of X-rays either with VNP20009 or ΔppGpp ST expressing cytolysin A (ClyA) or γ-radiation with *Salmonella* BRD509 induced a significant suppression of the tumor or delayed tumor growth [146,147,148]. In addition, the treatment with A1-R post-surgical excision of tumors significantly inhibited surgery-induced breast cancer metastasis [149]. In combination therapy, the use of prodrug strategy along with *Salmonella*-expressing, prodrug-activating enzymes such as HSV TK, carboxypeptidase G2 (CPG2), and cytosine deaminase have more promising tumor retardation capabilities compared to the use of the therapeutic strain alone [15,56,150].

## 11. Cancer Vaccines Delivered by *Salmonella*

Being an intracellular pathogen, with vast survival in different organs of the host, attenuated *Salmonella* has been widely used as a vaccine delivery system against various diseases [151]. As mentioned earlier, *Salmonella* is a multifaceted antagonist for cancer [152,153], apart from that, using auxotrophic *Salmonella* as a therapeutic and prophylactic vaccine delivery system is also an ideal strategy. Medina et al. have demonstrated the anti-cancer effect of auxotrophic ST (∆*aroA*) as a vaccine delivery system that expresses β-gal as a model TAA against aggressive fibrosarcoma [154]. In another study, SPI-2 and T3SS of *Salmonella* were used to deliver survivin as a TAA into antigen-presenting cells, and the PsifB::sseJ promoter/effector combination was found to have an excellent anti-cancer immune response of CD8 infiltration in the tumor environment [155]. Similarly, elevated effector-memory CTL responses against CT26 colon cancer and orthotopic delayed brain tumor glioblastoma in mice were found after immunization with survivin, which was fused to the SseF effector protein and kept under the regulation of SsrB, the key regulator of SPI2 [141]. Heat shock protein 70, as an immuno-chaperone fused with SopE of *Salmonella* T3SS, has elicited a considerable CTL response against murine melanoma [156]. A multi-antigen DNA vaccine encoding fusion antigenic domains of tyrosine hydroxylase, survivin, and PHOX2B, delivered by auxotrophic ST *(∆aroA*, *∆guaAB*), has been demonstrated to exhibit significant elicitation of the CTL response, INFγ production, and excellent suppression of neuroblastoma in a mouse model [157]. In addition to the CTL responses, the elicitation of humoral response by a *Salmonella*-based oral DNA vaccine with a MG7-Ag mimotope against gastric cancer was confirmed [158]. An H_2_O_2_-inactivated *S*. Typhimurium RE88 *(∆aroA*, *∆dam*) has been established to induce anti-cancer immunity by using ovalbumin as a model antigen [159]. *Salmonella* has also been studied for the expression of oncogenic virus antigens. A recombinant ST that produced Human Papillomavirus Type 16 (HPV16) L1 Virus-like Particles (VLPs) induced the anti-tumor immune response in prophylactic as well as therapeutic contexts [160]. The same group has constructed a *Salmonella* that has expressed major capsid protein L1 of HPV16 virus via plasmid, and has shown the induction of anti-HPV16 neutralizing and humoral immune responses [161]. They have also demonstrated the intravaginal immunization of the HPV16-L1 *Salmonella* construct and its innate, adaptive, Th1, and Th2 mucosal immune responses [162]. Thus, *Salmonella* can be used to deliver oncogenic viral antigens for prophylactic vaccine development. In addition, *Salmonella* infection triggers the formation of gap junctions in melanoma that are typically lacking in tumor cells. The transfer of tumor antigens to dendritic cells and the resultant induction of immune responses depend on these gap junctions [92]. Moreover, *Salmonella* has the ability to induce MHC class I and II immune responses by delivering cancer-related antigens via bacterial surface and translocating the antigen or its gene to the antigen-presenting cells, respectively. *Salmonella* has the virtue of being used as a delivery vehicle for extrinsic cancer antigens, oncogenic viral antigens, and to display the intrinsic antigens of the active tumor to achieve anti-cancer immunity based on these aspects.

## 12. Application of *Salmonella* in Tumor Targeting and Detection

Tumor targeting and accumulation phenotypes have made *Salmonella* the best player in the creation of genetically engineered strains for detecting tumors. Strains expressing fluorescent proteins are well studied for the purpose of visualizing and locating the tumor region in vivo [16]. Another approach for positioning tumor-specific *Salmonella* used positron emission tomography which locates the engineered ST VNP20009 strain in tumors by expressing the HSV1-TK reporter gene that can selectively phosphorylate radiolabeled 2′-Fluro-1-β-D-arabinofuranosyl-5-iodo-uracil [163].

*Salmonella* was engineered to express the fluorescent protein ZsGreen. It has a high sensitivity that can detect tumors 2600 times smaller than the current limit of tomographic techniques [164]. Since *Salmonella* preferentially accumulates in tumors and microscopic metastases, this approach would provide a method to detect a tumor, monitor treatment efficacy, and identify metastatic onset.

## 13. Clinical Trials

The success of attenuated ST strain VNP20009 with chromosomal deletion of the *purI* and *msbB* genes in order to target tumor and inhibit its growth in mice takes the lead for a Phase I human clinical trial in metastatic cancer patients through intravenous infusion [33]. In this study, none of the patients experienced tumor regression, although VNP20009 could be safely administered to patients. Tumor localization was noted at the highest tolerated dose in only 3 patients [33]. Further improvement on the VNP20009 was incorporated by inserting a gene encoding *E. coli* cytosine deaminase (CD) and designated the strain as TAPET-CD [165]. Three patients received an intratumoral injection of TAPET-CD; two patients had intratumor evidence of bacterial colonization that persisted for at least 15 days after the initial injection. The conversion of 5-fluorocytosine to 5-fluorouracil as a result of CD expression was demonstrated in these two patients with a tumor-to-plasma ratio of 3:1. This signifies the localization of TAPET-CD in tumors with minimal systemic spread. In another study, oral vaccination with live attenuated *Salmonella typhi,* carrying an expression plasmid encoding VEGFR2, increased vaccine-specific T cell responses in patients with advanced pancreatic cancer [166]. Aside from human clinical trials, VNP20009 has been administered in a Phase I trial on dogs. The study revealed bacterial tumor localization in 42% of cases and major anti-tumor response in 15% of cases [70]. The exciting anti-tumor activity in tumor-bearing dogs or mice models was not reflected in human clinical trials. This discrepancy in anti-tumor effects of engineered ST in animal and human models demand an in-depth study on the anti-tumor mechanism, especially in humans. Human clinical trials for anti-cancer treatment using *Salmonella* are listed in Table 2.

## 14. Limitations of Bacteria-Mediated Cancer Therapy

The race for bacteria-mediated tumor therapy is at its peak, as it holds potential for the treatment of different tumors. The main concern while using bacterial vaccines is their ubiquitous nature. Since many bacteria, including *Salmonella* and *Listeria,* used in BMCT are prevalent in the environment, and their pre-exposure commonly induces immunity against these pathogens. Therefore, the preexisting or vaccine-induced vector-specific immunity could potentially block therapeutic genes and vaccine delivery. In a survey, it was found that there was a preexisting cellular immunity to *L. monocytogenes* in 60% of human donors tested [168]. The current practice of genetic modification by deletion of genes responsible for the biosynthesis of lipopolysaccharide, amino acids, or purines affects bacterial virulence that modulates bacterial invasion, tumor localization, and immune stimulation. Excessive attenuation downregulates invasive potential, while less attenuation is pathogenic. Thus, balanced attenuation is a major concern for the development of effective anti-tumor ST strains, since insufficient attenuation might result in proliferation at non-designated organs, and may even cause severe septic shock while excessive attenuation affects its invasion potential and immunogenic presentation. While administering ST for treatment, termination of bacteria from the treated patient is another issue that needs to be resolved. Although antibiotics are the remedy, there is a high chance of acquiring antibiotic resistance. So far, bacterial application for cancer therapy needs to address safety issues despite having a promising success rate. SMCT is still facing many problems, and as a result, most phase I clinical trials have not been satisfactory. Furthermore, intrinsic factors such as tumor architecture, growth rate, and blood supply, as well as extrinsic factors, especially the entry of bacteria into tumors, their growth within tumors, and clearance from the peripheral circulation and tumors, might have a pivotal role in the success of SMCT in clinical application. Therefore, the complicated interactive mechanism between *Salmonella*, tumor cells, host immune cells, and inflammatory reactions needs maximum exploration to understand the process so that SMCT could be applied to human cancer therapy.

## 15. Future Perspectives and Conclusions

In this review, we elaborated on the anti-cancer therapeutic potential of *Salmonella*. Special features such as tumor-targeting capacity, tumor tissue penetration, immunomodulatory effects, genomic engineering plasticity, and delivery efficacy highlighted *Salmonella* as a genuine candidate for cancer therapy. Current knowledge was elaborated on in order to facilitate the development of safer and more effective *Salmonella* bacterial strains for cancer treatment. Although the anti-tumor potential of *Salmonella* has been elucidated, the precise mechanism of immune activation against cancer by *Salmonella* is still not clear. Even though the anti-tumor potential of *Salmonella* has been established in murine models and dog phase trials, clinical trials in human patients have not been effective. The success rate of tumor suppression by *Salmonella* in humans is still inadequate. Thus, *Salmonella*-based human oncotherapy still demands immense immunological and molecular mechanism studies on human prospects. Cancer patients and animals are highly immuno-compromised; thus, even mild systemic bacterial infection may lead to serious adverse effects. Hence, avirulent, non-endotoxic *Salmonella* strains are highly envisaged. Moreover, *Salmonella*-based oncotherapy requires consecutive inoculation. Upon subsequent inoculation, *Salmonella* should not be nullified by host immunity. Therefore, strains which are less immunogenic but highly invasive to tumors have a good scope. Genetic manipulation of *Salmonella* by simple genetic engineering or synthetic bioengineering is well known to induce the production of anti-cancer drugs. Furthermore, BMCT is the proof of concept that can either be used as a monotherapy or in combination with other anti-cancer therapies in order to achieve better clinical outcomes. The ongoing continuous development and innovation of *Salmonella*-based cancer therapy possesses great promise for a new paradigm of cancer therapy that could pave the way for its succeeding emergence as one of the mainstream therapeutic interventions to tackle various types of cancer.

## Figures and Tables

**Figure 1 pharmaceutics-14-02100-f001:**
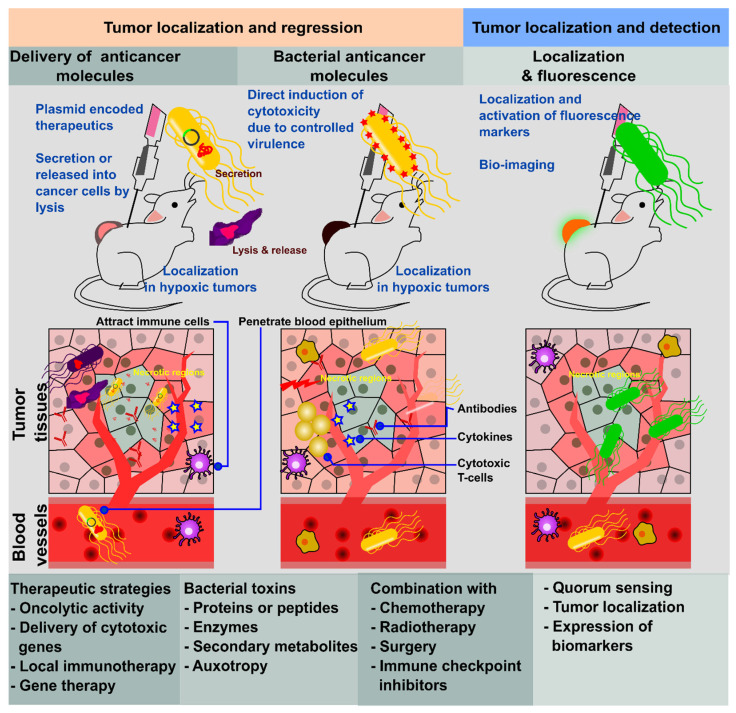
Therapeutic strategy for *Salmonella*-mediated cancer therapy. *Salmonella* has been exploited to localize and regress tumors via the delivery of anti-cancer molecules to induce cytotoxicity. The intrinsic tumor-targeting feature of *Salmonella* has been hijacked to deliver the gene of interest or secondary metabolites to suppress the tumor. Recently, combination therapy with conventional treatment strategies is more effective in the treatment of various cancers. Another promising approach is the bio-imaging of the tumor by using a fluorescence marker that enhances the localization and helps in targeting the metastatic tumor.

**Figure 2 pharmaceutics-14-02100-f002:**
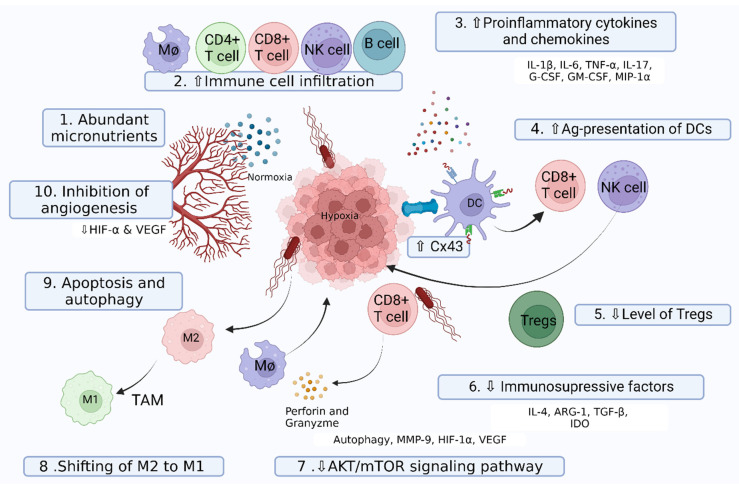
Schematic diagram illustrating the mechanism of *Salmonella* to manipulate host immune response for tumor inhibition. Transformation of the tumor microenvironment from immunosuppressive to immunogenic occurs through increased infiltration and reprogramming of anti-tumor immune cells, upregulating the expression of proinflammatory cytokines and inducing a shift in the phenotypic and functional characteristics of immune cells. (1) A niche full of micronutrients supports the aggressive growth of *Salmonella* [87]. (2) *Salmonella* invasion increases the infiltration of macrophages [88], CD4+ T cells, CD8+ T cells, NK cells [89], and B cells [90] in the tumor microenvironment. (3) Bacterial invasion upregulates the expression of proinflammatory cytokines [91]. (4) *Salmonella* infection upregulates the Cx43 protein to enhance an antigenic presentation by dendritic cells [92]. (5) Activation of CD8+ T cells and NK cells downregulates Treg cells [93,94]. (6) Downregulation of immunosuppressive factors favors better immune protective response [95,96]. (7) Cytotoxic T cells release perforin to lyse the tumor cells [38,97]. (8) Transition of M2 macrophage to M1 macrophage orchestrates the protective anti-tumor immune response [88,98]. (9) *Salmonella* induces apoptosis and autophagy through its intrinsic properties [99,100,101]. (10) Chaotic vasculature is disrupted by inhibition of angiogenesis via downregulation of hypoxia-inducible factor-1 alpha (HIF-1α) and vascular endothelial growth factor (VEGF) [38]. All of the steps are summarized in the text. MΦ: macrophage; IL: interleukin, G-CSF: granulocyte colony-stimulating factor; GM-CSF: granulocyte-macrophage colony-stimulating factor; MIP-1α: macrophage inflammatory protein-1 alpha; ARG-1: arginase-1; TGF-β: transforming growth factor-beta; IDO: Indoleamine 2,3-dioxygenase; TAMs: tumor-associated macrophages; DC: dendritic cell; M1: M1-like macrophage; M2: M2-like macrophage.

**Table 1 pharmaceutics-14-02100-t001:** *Salmonella* strains in therapeutic application in cancer.

Bacteria (Strain)	Strategy/Gene of Interest	Tumor Model	Results/Mode of Action	References
*S. Typhimurium* VNP20009 TAPET-CD	Cytosine deaminase expression from *E. coli* that converts non-toxic 5-fluorocytosine to the active anti-tumor agent 5-fluorouracil	Mice colon tumors	Mice treated with 5-FC inhibited tumor growth by 88−96% compared to TAPET-CD alone, which inhibited tumor growth by 38−79%	[55]
*S. Typhimurium* VNP20009	Carboxypeptidase G2 (CPG2)	Murine models of breast and colon cancer	Prodrug-based suicide gene therapy	[56]
*S. Typhimurium* VNP20009	*msbB* and *purI* mutations	melanomas	Attenuated strain preferentially accumulates in tumors and is rapidly cleared from other organs	[32]
*S. Typhimurium*	Deletion of ppGpp	CT26 tumor	Tumor suppression via IL-1β.	[50]
*S. Typhimurium* A1-R	leucine-arginine auxotroph	Pancreatic cancer orthotopic mouse model	Promotes CD8+ T cell infiltration and arrests tumor growth and metastasis.	[57]
*S. Typhimurium* A1-R	Combination with recombinant methioninase or doxorubin or temozolomide	Osteosarcoma, sarcoma, melanoma	Eradicate osteosarcoma and soft tissue sarcoma; regresses malignant melanoma	[58,59,60]
*S. Typhimurium* A1-R	Combination with temozolomide or vemurafenib	Melanoma in patient-derived orthotopic xenograft (PDOX) model	Combinatorial anti-tumor effect and drugs promoted targeting of S*. Typhimurium* A1-R	[61]
*S. Typhimurium* (ST2514P3)	Tryptophan auxotroph (*trpA trpE* deletion)	Breast cancer (4T1)	Suppressed the primary tumor growth and pulmonary metastasis	[13]
*S*. *Typhimurium*	Vascular endothelial growth factor receptor 2	Lewis lung carcinoma	Tumor suppression and inhibition of pulmonary metastasis	[62]
*S. Typhimurium* VNP20009	TNF-related apoptosis-inducing ligand (TRAIL)	Mammary tumor, melanoma	Caspase-3-mediated apoptosis in cancer cells	[54,63]
*S. Typhimurium*	Fas ligand (FasL)	breast carcinoma and CT-26 colon carcinoma cells	Tumor growth inhibition by 59% for breast tumors and 82% for colon carcinoma	[53]
*S. Typhimurium*	shRNA- expressing vectors targeting bcl2	Melanoma	Delayed tumor growth and prolonged survival	[64]
*S. Typhimurium*	Cytotoxic protein (HlyE)	Mammary tumor	Increased tumor necrosis and reduced tumor growth	[65]
*S. Typhimurium msbB- and purI-*	IL-18	Lewis lung carcinoma	Inhibit the growth of primary subcutaneous tumors and pulmonary metastases	[66]
*S. Typhimurium*	Herpes simplex virus thymidine kinase (HSV TK)	Melanoma	Suppressed tumor growth and a prolonged average survival	[15]

**Table 2 pharmaceutics-14-02100-t002:** List of *Salmonella* strains under human clinical trials for cancer therapy.

Bacteria	Strain	Cancer Model	Result	Phase	References
*S. Typhimurium*	VNP20009	Metastatic melanoma and renal cell carcinoma	Focal tumor colonization was recorded without tumor regression	I	[33,167]
*S. Typhimurium*	VNP20009 expressing TAPET-CD (Cytosine deaminase)	Squamous cell carcinoma and adenocarcinoma	Intratumoral bacterial colonization in 2 out of 3 patients	I	[165]
*S. Typhimurium*	VNP20009	Solid tumors	Not provided	I	https://clinicaltrials.gov/ct2/show/NCT00006254 (accessed on 1 September 2022)
*Salmonella typhi* (Express VEGFR2)	Ty21a (VXM01)	Pancreatic cancer	VXM01 vaccination increased vaccine-specific T cell responses	I	[166]

## Data Availability

Not applicable.

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
