# Peer review of "Salmonella as a Promising Curative Tool against Cancer"

_pharmaceutics, 2022, doi:10.3390/pharmaceutics14102100_

Round 1

Reviewer 1 Report

The review article presented by Aganja et al provides a good overview about use of Salmonella in cancer management. Though several other articles are available on this topic, the current version provides updation of text. The manuscript may be accepted for publication provided authors address following issues for the ease of clarity.

Line 102-104: Some bacteria mentioned in the list are also responsible for promotion of carcinogenesis. Therefore the text must be updated

The role of bacterial effectors involved in killing tumor cells from native bacteria must be discussed.

Point 3 (Line 178): Contradicting evidences about role of Salmonella in prevention and promotion of natural bacteria exist, but they are lacking here. This must be incorporated to give overall overview.

Bacterial nomenclature must be corrected in multiple instances, for example, E. coli in Table 1.

Figure2 presents several mechanisms for the antitumor activity of Salmonella. It will be better if the author provides relevant references for further study in the figure/caption

Author Response

Please see the attachment. (Response in blue color)

Reviewer 2 Report

This review by Ram Prasad Aganja et al summarizes the mechanisms, strategies, and potential engineering methods of Salmonella as a key microbial agent suitable for the next generation of cancer therapy. Salmonella Typhimurium is one of the most promising mediators, with engineering plasticity and cancer specificity. It can be used to deliver toxins that induce cell death in cancer cells specifically, and as a cancer-specific instrument for immunotherapy by delivering tumor antigens and exposing the tumor environment to the host immune system. Overall, the object reviewed in this manuscript is a hot topic in the field. Appropriate improvements are needed to be published in Pharmaceutics. Here are some comments:

1.     This review attempts to summarize the recent advancements in Salmonella-mediated oncotherapy in the fight against cancer, while the discussion about research for clinical trials is inadequate.

2.     There is little discussion about prophylactic effect of Salmonella against cancer in the review. Thus, the title “prophylactic tool against cancer” seems inappropriate.

3.     The subject of this review is Salmonella, so there seems no need to lengthily discuss the native and attenuated forms of other bacteria and viruses. On the contrary, it would be better to expand the discussion to features of native and attenuated forms of Salmonella for cancer therapy.

4.     “Hypoxic environment and tumor vasculature” is the characteristic of tumor tissues, which is inappropriately discussed as a separate chapter in the section “Anti-tumor mechanism of Salmonella”.

5.     The part “Orchestration of TAM function and polarization” focuses on introducing the function of TAMs, while the theme is the effect of Salmonella on TAMs.

6.     There are various limitations of Salmonella-mediated cancer therapy. The authors should discuss more about recent advances in methods of improving Salmonella-mediated oncotherapy, such as Nat. Commun., 2021, 12:6116; Nano Today, 2022, 45, 101537; etc.

7.     This research field has been extensively studied. The authors should discuss more insightfully and comprehensively, especially the perspectives and future directions of the field.

Author Response

Please see the attachment. (Response is in blue color)

Reviewer 3 Report

The paper is well written and constructed, and clear stated the backgroup and potential of the Salmonella in cancer treatment. Some small mistakes, such as the Italic, superscript can be corrected by the editors.

I suggest authors detect the sentence “ With this background, 21 Salmonella-mediated cancer therapy has achieved U.S. Food and Drug Administration approval for 22 clinical trials.”, as I remembered that this trail is failed for a long time ago.

In general, Ram et al discuss the history of the Salmonella in cancer treatments, and fully discuused different Salmonella strain, their advantages and limitations in this field, and the mechanisms. In addition, authors also discuss the Salmonella as a vector for protein drugs, which is the hot topic now.

I think this paper is well written, and systemly introduce many knowlge on Salmonella in cancer treatments, which is a good summary on this topic.

Author Response

(The authors gave the same response as above.)

Round 2

Reviewer 2 Report

No further comments.

Author Response

Since the Reviewer 2 commented as "No further comments" the manuscript has been submitted as such and considered the comments from editor for title change. The title has been changed to "Salmonella as a promising curative tool against cancer"